# Substance Use in the Transgender Population: A Meta-Analysis

**DOI:** 10.3390/brainsci12030366

**Published:** 2022-03-10

**Authors:** Miriam Cotaina, Marc Peraire, Mireia Boscá, Iván Echeverria, Ana Benito, Gonzalo Haro

**Affiliations:** 1TXP Research Group, Universidad Cardenal Herrera-CEU, CEU Universities, 12006 Castellon de la plana, Spain; miriam.cotaina@gmail.com (M.C.); mperaire@hotmail.com (M.P.); gomechiva@alumnos.uchceu.es (I.E.); gonzalo.haro@uchceu.es (G.H.); 2Department of Mental Health, Consorcio Hospitalario Provincial de Castellón, 12002 Castellon de la plana, Spain; mireia.bmart@hotmail.com; 3Torrente Mental Health Unit, Hospital General de Valencia, 46900 Torrente, Spain

**Keywords:** transgender, cisgender, substance use, gender differences, substance use disorders, addictions, meta-analysis

## Abstract

(1) Background: This meta-analysis aimed to assess the relationship between identifying as transgender and substance use. (2) Methods: We searched for relevant studies in PubMed, Scopus, the Web of Science, and PsycINFO on 21 July 2021. (3) Results: Twenty studies comparing transgender and cisgender people were included in this work, accounting for a total of 2,376,951 participants (18,329 of whom were transgender). These articles included data on current tobacco use, current tobacco use disorder, current alcohol use, current alcohol use disorder, lifetime substance (all) use, current substance use (excluding tobacco and alcohol), current use of specific substances (excluding tobacco and alcohol and including cocaine, amphetamines, methamphetamines, ecstasy, stimulants, heroin, opiates, cannabis, marijuana, LSD, hallucinogens, steroids, inhalants, sedatives, Ritalin or Adderall, diet pills, cold medicine, prescription medications, polysubstance, other club drugs, and other illegal drugs), and current substance use disorder (excluding tobacco and alcohol). We used the ORs and their 95% CIs to state the association between identifying as transgender and those variables. The control reference category used in all cases was cisgender. We employed a random-effects model. Transgender people were more likely to use tobacco (odds ratio (OR) = 1.65; 95% CI [1.37, 1.98]), have used substances throughout their lives (OR = 1.48; 95% CI [1.30, 1.68]), and present current use of specific substances (OR = 1.79; 95% CI [1.54, 2.07]). When current alcohol and substance use in general and tobacco, alcohol, and substance use disorders specifically were considered, the likelihood did not differ from that of cisgender people. (4) Conclusions: The presence of substance use disorders did not differ between transgender and cisgender people. Considering this population as consumers or as addicted may be a prejudice that perpetuates stigma. Nonetheless, transgender people were more likely to use tobacco and other substances, but not alcohol. Hypothetically, this might be an emotional regulation strategy, a maladaptive mechanism for coping with traumatic experiences, or could respond to minority stress, produced by stigma, prejudice, discrimination, and harassment. It is of particular importance to implement policies against discrimination and stigmatisation and to adapt prevention and treatment services so that they are inclusive of the 2SLGBTQIA+ community.

## 1. Introduction

The term transgender describes individuals whose gender identity or expression differs from the sex they were assigned at birth. Some people prefer to view these congenital conditions as a matter of diversity and use the terms intersex or intersexuality instead. This term includes not only people whose gender identity differs from the sex assigned at birth, but also those whose reproductive organs do not conform to what is traditionally designated as male or female. These terms encourage the conception of gender from a non-binary perspective of it. Transgender is an umbrella term preferred by many because it is more inclusive. The intersex concept must be differentiated from people who identify themselves as asexual, which represents those people who typically do not experience sexual attraction or want to pursue sexual relationships with other people. Otherwise, people who identify with the sex assigned to them at birth are considered cisgender, meaning that their gender identity aligns with their biological sex and cultural roles. Likewise, within gender minorities, there is also people known as genderqueer, agender, non-binary, two-spirit, or gender fluid, which includes people whose gender identity and/or role does not conform to a binary understanding of gender as something limited to the categories of man or woman and masculine or feminine [1].

There are large differences in the prevalence of transgender individuals in different countries, with estimates varying between 0.40 [2] and 23.6 [3] per 100,000 inhabitants, even within regions of the same country [4,5,6]. Regarding the disaggregated prevalence of transgender people, figures are estimated at between 1:11,900 and 1:45,000 inhabitants for male-to-female and 1:30,400 and 1:200,000 for female-to-male people [1]. 

Transgender individuals can experience intense stigma along with social exclusion and marginalisation [1]. In addition, the risk of physical and sexual victimisation is significantly higher among this population with respect to the cisgender demographic, while the lack of legal protection against discrimination fosters further vulnerability in this group [7]. This context can favour risky behaviours, motivated by psychological discomfort, which then becomes chronic [8]. In this sense, and as is the case in the rest of the lesbian, gay, bisexual, transgender, queer, intersex, and asexual (2SLGBTQIA+) community, this population could present patterns of substance abuse different from those that occur in the heterosexual population [9], especially by being exposed to a greater risk of consuming tobacco, alcohol, or other drugs [10]. 

Although some studies consider the risk of substance abuse within the 2SLGBTQIA+ community to be similar to that of cisgendered people [11,12,13], others conclude that the risk is even greater for transgender individuals [14], in particular highlighting alcohol consumption among this population [15]. For some subjects, the use of this substance could constitute an avoidant coping skill used to escape from negative emotions [16,17,18,19,20] and mitigate the stress produced by structural and internalised transphobia, as well as identity concealment [15]. Bars with an 2SLGBTQIA+ atmosphere constitute one of the few spaces perceived by some subjects of this community as safe to meet and socialise without fear of discrimination, which could increase recreational substance use [21] because these places constitute environments that normalise consumption [22].

There is evidence to support the minority stress model, but in the case of the transgender population, it might still be scarce, as much of the research has focused on transgender women with multiple intersectional disadvantages [23]. In this way, research studying the incidence, aetiology, and peculiarities of substance use in the transgender population is not common [9,11] and usually presents small samples that refer only to the subgroup of the transsexual population (people whose gender identity does not coincide with their biological sex and who wish to undergo a hormonal and/or surgical transition towards gender affirmation) [1,4,5,6], thereby making it difficult to understand the unique characteristics of this group. Moreover, these studies are very heterogeneous in terms of their definitions and analysis of the samples under study, meaning that it is common for individuals with diverse sexual identities to be grouped indistinctly together alongside other individuals with a minority sexual orientation [1,21,24,25]. In these cases, the distinctive characteristics of the transgender community are not specifically considered, and so, there is a risk of minimising the problems this population faces [11].

Therefore, this meta-analysis arose from our interest in addressing the knowledge gap regarding the use and abuse of substances by transgender people. Our objective was to group and analyse the results published to date in order to quantify the probability (pooled odds ratio) of presenting use and suffering from a substance use disorder among transgender people compared to the cisgender population, in other words to compare the prevalence of substance use and substance use disorder in transgender and cisgender people.

## 2. Materials and Methods

### 2.1. Protocol and Registration

This report was prepared according to the PRISMA guidelines for reporting in systematic reviews and meta-analyses [26]. The protocol was registered with the Prospero Centre for Reviews and Dissemination on 29 September 2021 (CRD42021275165): Transgender and substance use: a meta-analysis. Available online: http://www.crd.york.ac.uk/prospero/display_record.php?RecordID=275165 (accessed on 30 January 2022).

### 2.2. Search Strategy

We searched for relevant studies in PubMed, Scopus, the Web of Science, and PsycINFO on 21 July 2021. The search start year was not limited. The search terms used were “(Transgender OR Transsexual) AND (Substance use OR Addiction OR Substance abuse OR Substance use disorder OR Drug OR Drugs)”. We also reviewed the list of citations included in the articles, reviews, and meta-analyses.

### 2.3. Inclusion and Exclusion Criteria

The inclusion criteria were: (1) quantitative comparison studies regardless of the design, comparing the prevalence of substance use and disorders between transgender and cisgender subjects, whether they considered substances in general or specific substances; (2) individuals who identified themselves as transgender were considered; (3) the participants had been classified as transgender or cisgender; (4) the study outcomes were substance use, consumption, abuse, dependence, or addiction; (5) the original papers reported odds ratios (OR) and their 95% confidence intervals (CIs) for substance use, consumption, abuse, dependence, or addiction (or data that could be used to calculate these). Relevant studies were included irrespective of their publication language, date of publication, or the nationality, race, age, or sexual orientation of the participants considered.

The exclusion criteria were studies that: (1) did not include a comparison group or the comparison was a narrative/qualitative; (2) did not distinguish between transgender and diverse sexual and gender orientations and identities (if the studies differentiated between transgender and non-binary, only the group that identified as transgender was included); (3) the outcome was age of onset or the quantity or frequency of consumption; (4) mixed outcomes (substance use and mental disorders) were reported, from which the odds ratio of substance use could not be separated.

### 2.4. Data Extraction 

The articles assessed for eligibility were divided into two equal parts; two authors independently extracted the information from each half of them (a total of four researchers) using a standardised form and resolving any disagreements by discussing the matter with the other two authors until a consensus was reached. The variables extracted from the studies were author names, year, country, language, study population, sample size, sample age, sex assigned at birth, type of substance and substance use outcome (current use, substance use disorder, or lifetime use), study quality evaluated using the Newcastle–Ottawa Scale (NOS) [27], ORs, and related 95% confidence intervals (95% CIs; preferably with the most adjusted factors). When data for the meta-analysis were not displayed directly, they were calculated using the published data or with data requested and provided by the publication authors. 

### 2.5. Summary Measures 

We used the ORs and their 95% CIs to state the association between identifying as transgender and current substance use (if the subject currently uses the substance, if the subject has used it in the last 15–30 d), current substance use disorder (if the subject was diagnosed with a substance use disorder, including both abuse (the substance is consumed despite the problems and negative consequences it causes) and dependence (substance use causing tolerance, withdrawal, and/or a pattern of compulsive use)), and lifetime substance use (if the subject has ever consumed the substance throughout life). The control reference category used in all cases was cisgender. We employed a random-effects model weighting the studies by the inverse of the variance and performed all the statistical analyses using Epidat 3.1 software (Xunta de Galicia, A Coruña, Spain; Pan American Health Organization, Washington, DC, USA). Because this program was designed in Spain, the decimals in its outputs are expressed with commas. Therefore, when they appear later in the text, in Figures 2–6 and Appendix A, commas should be interpreted as decimal points.

### 2.6. Quality Assessment 

The NOS [27] was used to evaluate the methodological quality of the studies included. This scale allocates a maximum of 9 stars to the following domains: selection, comparability, and exposure. Two of the investigators independently performed the quality analysis, then compared the results until consensus was reached.

### 2.7. Heterogeneity and Publication Bias

Heterogeneity was evaluated using DerSimonian–Laird Q tests with Galbraith graphics. Possible publication bias was examined by employing Egger and Begg tests with funnel plots. 

### 2.8. Sensitivity and Subgroup Analysis 

We performed a sensitivity analysis by repeating the meta-analysis the same number of times as the selected datasets, each time omitting one dataset and combining all the remaining ones and then plotting the influence graphs. We decided whether we needed to analyse subgroups of studies by performing heterogeneity and sensitivity analyses. Specifically, the meta-analyses were repeated, eliminating the articles that contributed the most to heterogeneity according to the Galbraith graph.

## 3. Results

### 3.1. Studies Included

Our initial search returned 7083 article hits, and we eventually included 20 articles in this meta-analysis; Figure 1 shows the evaluation process we followed to select these studies. These studies included a total of 2,376,951 participants, of whom 18,329 identified as transgender.

Of the 20 selected articles, there were data on current tobacco use in 14, current tobacco use disorder in 3, current alcohol use in 9, current alcohol use disorder in 10, lifetime substance use (including tobacco, alcohol, and other substances) in 19, current substance use (excluding tobacco and alcohol) in 5, current use of specific substances (excluding tobacco and alcohol and including cocaine, amphetamines, methamphetamines, ecstasy, stimulants, heroin, opiates, cannabis, marijuana, LSD, hallucinogens, steroids, inhalants, sedatives, Ritalin or Adderall, diet pills, cold medicine, prescription medications, polysubstance, other club drugs, and other illegal drugs) in 40, and current substance use disorder (excluding tobacco and alcohol) in 4. Although the results obtained regarding current tobacco use disorder, current substance use, and current substance use disorder were unreliable because we found so few studies examining these topics, we included them in the Appendix A for informational purposes. A meta-analysis was completed for the relationship between identifying as transgender and substance types and use (current use, current use disorder, and lifetime use). Table 1 shows the characteristics of the studies included in these meta-analyses [12,19,28,29,30,31,32,33,34,35,36,37,38,39,40,41,42,43,44,45], which were all in English and included both sexes assigned at birth: male and female.

### 3.2. Current Tobacco Use and Tobacco Use Disorder

As shown in Figure 2, the pooled OR of presenting current tobacco use (*n* = 14) was 1.65 (95% CI [1.37, 1.98]) for transgender people (the cisgender/transgender ratio was 1:1.65); the Q index was 78.38 (*p* < 0.001). Both the Q index and the Galbraith graph indicated the presence of data heterogeneity. The data from De Pedro (2017) [35], Gamarel (2020) [38], and Azagba (2019) [29] for cigarettes and Kelly (2015) [42] and Dinger (2020) [36] for e-cigarettes contributed the most heterogeneity to this current study. Despite this heterogeneity, our sensitivity analysis showed that by removing each of these articles, the OR changed very little, and the precision did not increase (Figure 2). 

Nevertheless, we also performed a subgroup analysis excluding all these articles, thereby removing the data heterogeneity (Q = 7.62; *p* = 0.470), which resulted in a slight reduction in the pooled OR of 1.58 (95% CI [1.44, 1.73]). Regarding publication bias, as also shown in Figure 2, the Berg test result was Z = 0.21 (*p* = 0.826) and the Egger test result was t = −0.02 (*p* = 0.977). These results indicate that there was no publication bias since the dispersion could be attributed to heterogeneity. 

The pooled OR of presenting current tobacco use disorder (*n* = 3) was 1.52 (95% CI [0.94, 2.45]), and the Q index was 13.05 (*p* = 0.001); the forest plot, Galbraith and influence graphics, and the funnel plot are shown in Appendix A.

### 3.3. Current Alcohol Use and Alcohol Use Disorder

The pooled OR of presenting current alcohol use (*n* = 9) was 0.97 (95% CI [0.83, 1.14]) for transgender people (Figure 3). In particular, the inaccuracy (CI width) of the data from Kelly (2015) stood out to us. In this sense, as shown in Figure 3, the Q index was 34.90 (*p* < 0.001), which along with the Galbraith graphic, indicated the presence of heterogeneity. The data from Aparicio-García (2018) [28], De Pedro (2017) [35], and Dinger (2020) [36] contributed most of the heterogeneity. Sensitivity analysis showed that removing each of these items changed the OR very little, although a subgroup analysis in which these three items were excluded to remove this heterogeneity (Q = 5.90; *p* = 0.315) resulted in a slightly higher pooled OR of 1.03 (95% CI [0.93, 1.15]). Regarding publication bias, the Berg test result was Z = 0.72 (*p* = 0.465) and the Egger test result was t = −1.44 (*p* = 0.190), indicating that there was no publication bias (Figure 3).

As shown in Figure 4, the pooled OR of presenting current alcohol use disorder (*n* = 10) was 1.09 (95% CI [0.80, 1.49]), and the Q index was 170.67 (*p* < 0.001) for transgender people. Both the Q index and the Galbraith graphs indicated the presence of data heterogeneity. Data from De Pedro (2017) [35], Batchelder (2021) [30], Jun (2019) [41], Day (2017) [12], Stanton (2021) [44], and Dinger (2020) [36] contributed the most to the data heterogeneity, although sensitivity analysis showed that removing each of these items changed the OR very little (Figure 4). Moreover, a subgroup analysis in which these items were excluded to remove the heterogeneity (Q = 2.86; *p* = 0.412) did not substantially change the pooled OR of 1.01 (95% CI [0.69, 1.28]). Regarding publication bias, as also shown in Figure 4, the Berg test results were Z = 0.89 (*p* = 0.371), and for the Egger test, it was t = 0.23 (*p* = 0.819). Given that the dispersion could be attributed to heterogeneity, these results indicated that there was no publication bias.

### 3.4. Lifetime Use of Substances (Including Tobacco, Alcohol, and Other Drugs) 

As shown in Figure 5, the pooled OR for lifetime use of any substance (*n* = 19) was 1.48 (95% CI [1.30, 1.68]) for transgender people (the cisgender/transgender ratio was 1:1.48), while the Q index was 314.72 (*p* < 0.001), which combined with the Galbraith graph indicated the presence of heterogeneity. The data from De Pedro (2017) [35] and Hoffman (2018) [40] regarding tobacco, Aparicio-García (2018) [28] in relation to drug and alcohol use, and Day (2017) [12] about tobacco contributed most of this heterogeneity. Despite this, the sensitivity analysis showed that the OR changed very little after removing each of these datasets (Figure 5). Indeed, in a subgroup analysis that excluded these data, thereby removing the data heterogeneity (Q = 10.24; *p* = 0.114), the pooled OR remained largely unchanged at 1.38 (95% CI [1.28, 1.49]). As also shown in Figure 5, in terms of publication bias, the Berg test results were Z = 0.48 (*p* = 0.624), and the Egger test results were t = 1.55 (*p* = 0.139). Thus, given that the dispersion could be attributed to heterogeneity, these results indicated that there was no publication bias.

### 3.5. Current Substance (Excluding Tobacco and Alcohol) Use and Use Disorder

The pooled OR of presenting current substance use (excluding tobacco and alcohol; any other substance, without specifying; *n* = 5) was 1.12 (95% CI [0.58, 2.15]) for transgender people (the forest plot, Galbraith and influence graphics, and the funnel plot are shown in Appendix A).

The pooled OR of presenting current substance use when the other substances (except tobacco and alcohol) were individually considered (*n* = 40) was 1.79 (95% CI [1.54, 2.07]) for transgender people (the cisgender/transgender ratio was 1:1.79). In addition, the Q index was 245.71 (*p* < 0.001) and, when considered with the Galbraith graphic, showed high heterogeneity, which was expected given the variety of the substances included in the analysis. Thus, we excluded the data that contributed the most to this heterogeneity: all the data from De Pedro (2017) [35] and Carone (2020) [32] for drugs; Aparicio (2018) [28] for drugs; Flentje (2014) [37] for cocaine, marijuana, heroin, and other drugs; Cohan (2006) [33] for cocaine/crack and any illicit drugs; Dinger (2020) [36] for inhalants, other club drugs, and marijuana; Day (2017) [12] for other drugs; and Hawke (2021) [39] for intra-COVID substance use. As shown in Figure 6, when we performed the sub-analysis with the resulting subgroup, the heterogeneity was eliminated (Q = 31.95; *p* = 0.059), and the pooled OR increased to 2.11 (95% CI [1.77, 2.51]; the cisgender/transgender ratio was 1:2.11). The Berg test (Z = 1.08; *p* = 0.276), Egger test (t = −0.23; *p* = 0.818), and funnel plots indicated the absence of publication bias.

As shown in the forest plot in Appendix A, the pooled OR of presenting current substance use disorder (*n* = 4) was 1.53 (95% CI [0.91, 2.59]). The Galbraith and influence graphics and the funnel plot are shown in Appendix A.

### 3.6. Summary of the Results

Table 2 shows a summary of the results obtained in this study.

## 4. Discussion

The results of this study showed that, compared to the cisgender population, transgender individuals have a greater probability of current use of tobacco and specific substances and of having consumed any substance over their lifetimes. In relation to tobacco, we found that the use of cigarettes, cigars, or e-cigarettes was higher among transgender individuals compared to their cisgender peers [31]. In a study that included 350 transgender people, 64% reported having used tobacco, 23% had perceived their use as problematic at some point in their lives, and 13% believed their use was currently problematic [46]. Indeed, previous studies have highlighted the structural discrimination suffered by the transgender population as one of the factors motivating their increased tobacco consumption, suggesting that this consumption is related to the desire of this populace to reduce their perceived stress levels [47].

In addition to structural discrimination, part of the transgender group is also regularly exposed to situations of social exclusion, marginalisation, and sex work in which drug use (mainly tobacco use) is very frequent [48]; these situations are more likely to occur in the group of transgender women compared to transgender men. This led some studies to explore intragroup differences, which indicated a higher prevalence of tobacco use among transgender men compared to transgender women [31]. Other work described the opposite, with a higher risk of tobacco and other drug use found among the group of transgender women [49]. However, still other studies pointed towards heterogeneity between the general transgender population and the cisgender men group, with an increased prevalence of tobacco use among these aforementioned groups compared to the female cisgender group [48]. 

All these discrepancies mean that more intragroup studies of the transgender population compared to the cisgender population will be required to assess the association between the experience of discrimination and the use of tobacco. In our study, the prevalence of consumption was higher, while in the other study, the age of consumption onset was earlier in the transgender group, often in the early stages of adolescence [50]. These data, which point to greater vulnerability of the group as the result of the discrimination they are subjected to, tended to homogenise in adulthood because of the increased concern for health and a desire to stop consuming among this population [51]. Of note, when we considered tobacco use disorder in this current study, we found no differences between the transgender and cisgender groups, although the small number of studies included in this analysis meant that our confidence in the reliability of this result was poor. 

Regarding substance use, our results coincided with other studies indicating that transgender people have a higher prevalence of substance use than cisgender people [23,35,52]. Specifically, one study estimated that transgender students were twice as likely as their cisgender peers to use cocaine or amphetamines and three times as likely to abuse inhalants [35]. 

Another problem was the misuse of prescription drugs, which was associated with lower self-esteem, greater discrimination based on gender identity, and increased self-reported symptoms of anxiety, depression, and somatic distress [53]. Moreover, this misuse was more frequent in binary transgender men and non-binary individuals, compared to binary transgender women [54]. Indeed, 24% of transgender women reported misuse of prescribed medications, in particular highlighting the use of analgesics (21.2%), anxiolytics (14.4%), stimulants (12.5%), and sedatives (8.7%) [53]. 

The factor most strongly associated with abuse, especially in the case of prescription opioids and tranquilisers, was the age at which the first medication had been prescribed, with transgender students being twice as likely to abuse analgesics compared to their cisgender peers [35,55]. However, no differences were found in this study between transgender and cisgender people in terms of general substance use or substance use disorder. These results contradict previous studies showing that transgender people were more likely to suffer from a substance use disorder, specifically with amphetamines, cocaine, or cannabis [52,56,57]. Nonetheless, our analyses included a small number of studies; therefore, the results must be interpreted with caution.

Several hypotheses have been proposed about the reasons for this relationship between substance use and identifying as transgender. One of the most accepted of these is the minority stress theory [58,59,60], which considers that the stigma, prejudice, discrimination, and harassment these individuals regularly receive from society could favour the appearance of depression, anxiety, or suicidal ideation, given that the prevalence of these disorders is higher in this group compared to their cisgender peers [61,62,63]. Thus, the use of substances would arise as a response to this discomfort and as a product of both internal factors (self-stigma, expectations of rejection, and non-conformity with self-image) and external factors (interpersonal and structural discrimination) [23,64]. 

Furthermore, the scarcity of specific resources aimed at the transgender demographic, as a consequence of institutional discrimination, would act as a trigger for consumption in this context [65,66,67]. In this sense, other studies indicate that substance abuse is linked to symptoms of depression in transgender women [68], increased stigma [69], and being a victim of transphobic discrimination [70]. Moreover, the risk of violence victimisation has also been significantly associated with substance misuse [55], especially in students who identify themselves as transgender or male. [71].

Another hypothesis to explain the higher prevalence of substance abuse among transgender individuals considers consumption as an emotional regulation strategy or a maladaptive coping mechanism to deal with traumatic experiences [41,67]. Some studies concluded that the presence of trauma in the biography of transgender individuals is decisive, given that, for example, more than half of transgender women consumers report having experienced some type of trauma in their lives [69]. In fact, compared to the cisgender population, young people from sexual minorities are more likely to have been victims of child abuse [46].

It is also interesting to consider the impact of psychiatric comorbidities on illicit substance use as a possible moderator. Most papers included in our analysis did not explore this relationship, but there is evidence that transgender people suffer a high prevalence of mental health disorders, highlighting anxiety [72], depression, self-harm, and suicidal ideation [73]. Previous studies carried out in the cisgender population demonstrated the impact that psychiatric comorbidities have on substance abuse, increasing the severity of addiction and functional impairment [74]. Although there are no specific studies focused on dual disorders in transgender individuals, the prevalence of psychiatric comorbidities in this population led us to hypothesize that substance abuse could be modulated by these diseases, being more severe and requiring a comprehensive approach that integrates or improves coordination between substance abuse and mental health care systems [74].

The most unexpected result of this present study was that there were no differences in the probability of presenting alcohol use or alcohol use disorder between cisgender and transgender people. In contrast, previous studies have postulated that one of the toxic substances with the greatest presence among the transgender population is alcohol [46,51,56]. Indeed, 60.4% of this population reported consuming alcohol regularly, and 24.3% reported having drunk excessively in the last 30 d [70]. In addition, it has been shown that transmasculine people are more likely to report binge drinking compared to transfeminine individuals, with this finding being linked to sex work in both cases [75]. 

Among transgender women, binge drinking was significantly higher in those who used amphetamines, had depressive symptoms, dropped out of school because of their gender identity, or had experienced verbal abuse [8]. In this sense, those who experience sexual abuse are also three times more likely to use cocaine, and those who had made suicide attempts had a higher risk of using marijuana [69]. To explain this divergence, it is important to remember that alcohol is also one of the drugs with the greatest presence in general society [76]. For example, in the USA (where most of the studies included in this work had been conducted), more people aged over 12 y had used alcohol in the year prior than any other drug or tobacco product, and alcohol use disorder was the most common type of substance use disorder [77]. 

The fact that no differences in the probability of presenting alcohol use or alcohol use disorder were found between cisgender and transgender people could perhaps be explained because of (1) the factors described in the previous paragraph, (2) the large sample size of our study, and (3) the enormous disproportion between the transgender and cisgender groups in the studies we considered, with the latter meaning that transgender people accounted for less than 1% of our sample. Thus, it is possible that the difference in prevalence of each individual study would have been insufficient to reach statistical significance when the combined probabilities were calculated. Another reason for this discordant result could be that the statistical difference found in these studies was not clinically significant. A debatable issue is that we considered binge and heavy drinking as abuse, and therefore, we included it in alcohol use disorder. We did this because we considered that a consumption of this entity is practically impossible not to cause some negative consequence, so we differentiated it from the use of alcohol, which would be a moderate consumption limiting intake to two drinks or less in a day for men and one drink or less in a day for women.

The fact that several studies were in populations from sexual and gender minorities may have also influenced the results, meaning that the prevalences would have been more similar than if they had been compared to the general cisgender population. Regarding alcohol use disorder, many of the studies we considered included samples of young people in which it may have been too early for the disorder to have established. In spite of all the above, our favourite hypothesis is that as the rights of sexual and gender minorities advance and the stigma decreases, the prevalence of consumption will gradually equalise between cisgender and transgender people. 

As limitations of this study, the main is the heterogeneity between the studies we included with respect to their populations (students, general population, sexual and gender minorities, sexual workers), cohort age (a minimum of 29% of the sample was under the age of 18), consumption type (current use, lifetime use, abuse, and dependence) and how to evaluate it, and substances evaluated. The heterogeneity and sensitivity analysis and the low number of studies in some of the analysis and categories did not allow the analysis of specific subgroups, but the influence of variables such as age, sample type, and other identities is an open question. Likewise, the search terms used could have left studies referring to a specific substance out of the results. The fact that these terms were in English probably influenced that all the studies were in that language and, except for two, the rest were carried out in North America (most of them in the United States). Nonetheless, no publication bias was detected, and the sensitivity and subgroup analyses indicated that the results were robust. Another limitation was the small number of studies included in three of the analyses, which meant that we did not have reliable results regarding general substance use, tobacco use disorder, or substance use disorder. Furthermore, in some studies, we had to calculate the OR directly from the data, and so, this figure had not been adjusted for other relevant variables in these cases. However, we considered that the adjusted ORs would probably have been smaller than those we found, and so, the result regarding alcohol would have likely remained the same. The lifetime substance use analysis could be less informative due to the inclusion of tobacco and alcohol; however, the sensitivity analysis indicated that excluding these studies would not substantially change the result. Another limitation is that the quantity and frequency of use were not assessed.

Finally, the transgender and cisgender populations were directly compared without segregation, according to the sex assigned at birth, sexual orientation, or the binary–non-binary variable. We performed the analysis in this way to obtain a first quantitative synthesis of the difference between the transgender and cisgender populations and because, otherwise, even less data would have been available in some of our analyses. The availability of this quantitative synthesis is precisely the strong point of this current study. However, in order to obtain more information in this regard and perform intragroup analyses, as several of the articles included in this study did (29, 36–38, 40, 41, 44), future studies should always report segregated results. Moreover, it would also be interesting to promote studies in the general population specifically focused on the transgender population, not only as part of the 2SLGBTQIA+ collective.

Taking these limitations into account, our data indicated that, compared to cisgender people, transgender individuals had a higher risk of presenting tobacco and substance use, but they did not differ in terms of alcohol use or alcohol use disorder. Two conclusions follow from this. First, although transgender participants were more likely to use tobacco and substances, this probability was not much higher than that of cisgender individuals. Indeed, this difference reached a ratio of only 2:1 when the data that generated heterogeneity in the consumption of specific substances were eliminated. In addition, there were no differences for alcohol use or for any substance use disorder. Thus, we would like to think that, as the stigma suffered by this population progressively reduces, consumption will also decrease. In turn, this could reduce the prejudice and stigma of considering this population as people who use drugs and as “addicts”. 

Second, although our study did not test it, hypothetically, the greater probability of tobacco and substance use among transgender individuals may constitute an emotional regulation strategy or a maladaptive coping mechanism in the face of traumatic experiences. This strategy could also respond to minority stress caused by stigma, prejudice, discrimination, and bullying. However, the desire to reduce substance use was similar between transgender and cisgender people, with transgender individuals reporting a greater need for help and being more likely to seek it [78]. The problem is that detoxification and drug dependency treatment centres are designed to serve cisgender heterosexual populations, and so, apart from exposing themselves to stigmatic and prejudiced attitudes, transgender people also face heteronormative barriers in structural and programmatic elements [79]. 

Thus, specialised substance abuse interventions must take gender minorities into consideration, and the professionals delivering them must have sufficient training to provide this demographic with appropriate care by, for example, considering the influence of minority stress [65]. This is especially important for prevention, as our findings showed that transgender people do not have higher odds of substance use disorder, but rather are more likely to use substances, especially at younger ages (since much of the sample was middle/high school students). It is critical to address addiction prevention in relation to sexual identity, lived experiences, related stressors, social and cultural contexts [64], and risky sexual behaviours [56]. To do this, anti-discrimination and stigmatisation policies must be implemented, and substance abuse prevention and treatment services should be adapted to be inclusive of the 2SLGBTQIA+ community. More research will be required to explore the barriers to accessing specialised services that transgender individuals experience. This would allow researchers to understand the specific needs of this population and develop preventive interventions and adapted treatments [24,80]. To ensure that substance abuse treatment services are inclusive, it would perhaps be advisable to register gender identity and to design specific interventions for this demographic.

## 5. Conclusions

Transgender people do not differ from cisgender people in terms of alcohol use or in the presentation of substance use disorders. This indicates that considering the population of transgender individuals as consumers and addicts likely represents a stigma-generating prejudice. However, transgender people are more likely to use tobacco and other substances. Hypothetically, this consumption may constitute an emotional regulation strategy or a maladaptive coping mechanism in the face of traumatic experiences. This strategy could also respond to minority stress caused by stigma, prejudice, discrimination, and bullying. Therefore, policies against discrimination and stigmatisation must be implemented, and prevention and treatment services should be adapted to make them inclusive of the 2SLGBTQIA+ community.

## Figures and Tables

**Figure 1 brainsci-12-00366-f001:**
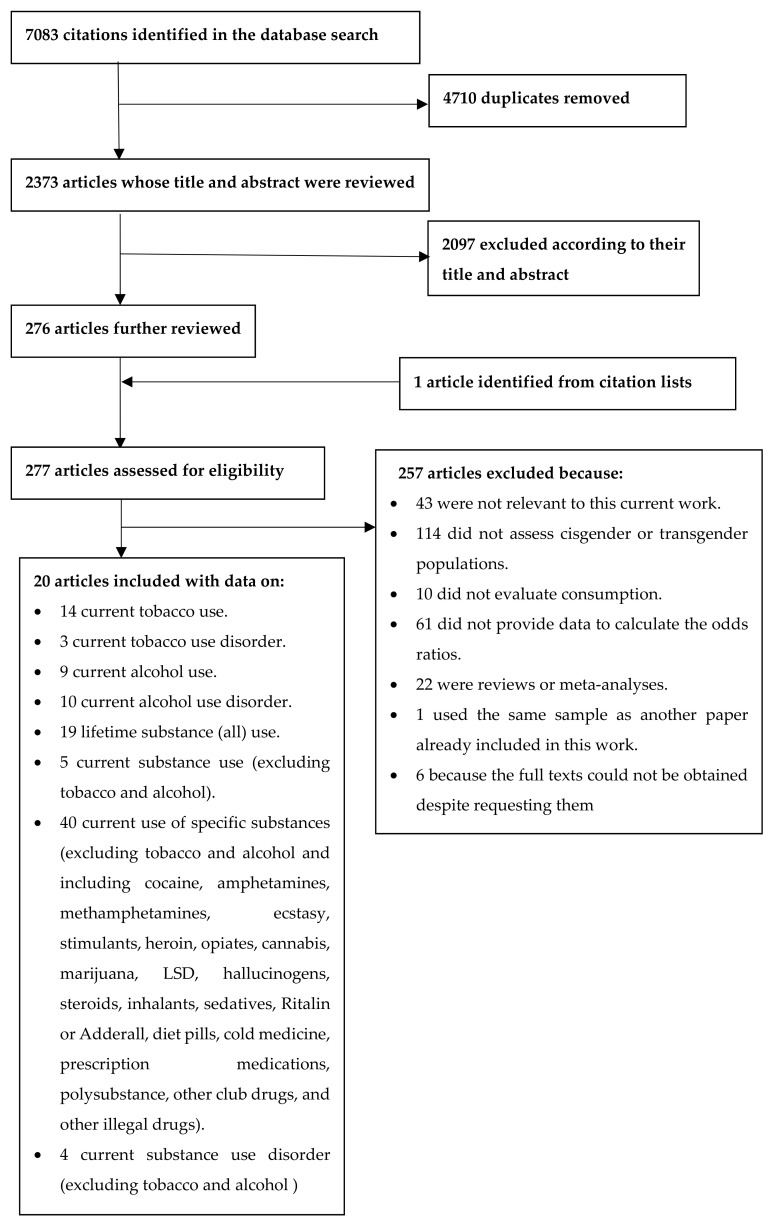
Flowchart of the systematic review process used to select articles for inclusion in this meta-analysis.

**Figure 2 brainsci-12-00366-f002:**
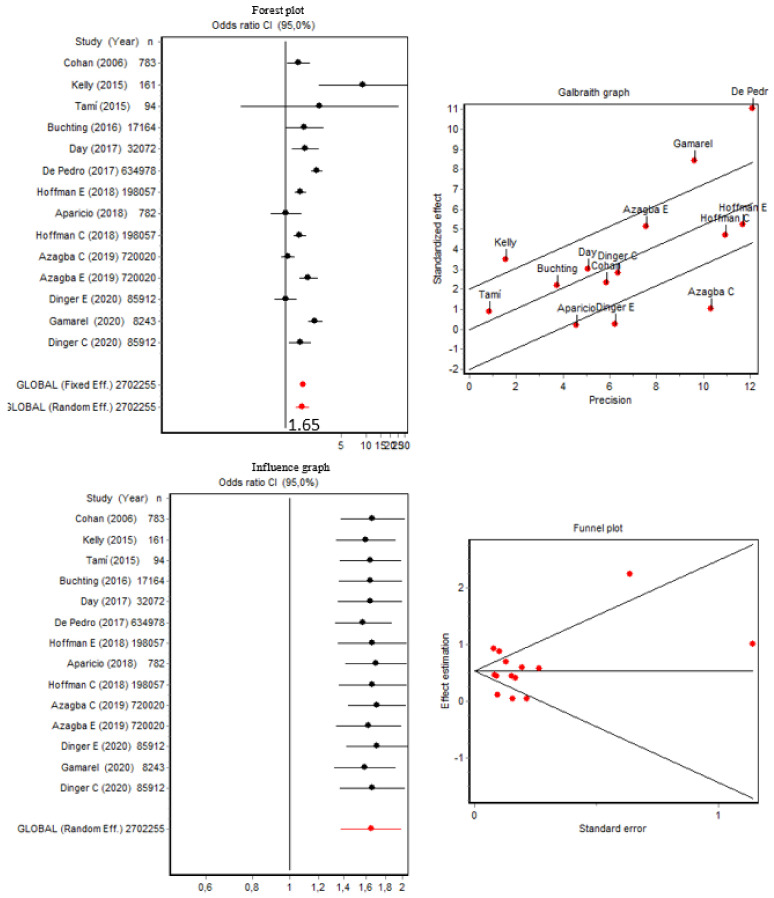
Forest plot, Galbraith graphic, influence graphic, and funnel plot for current tobacco use. Abbreviations: C, cigarettes; E, e-cigarettes.

**Figure 3 brainsci-12-00366-f003:**
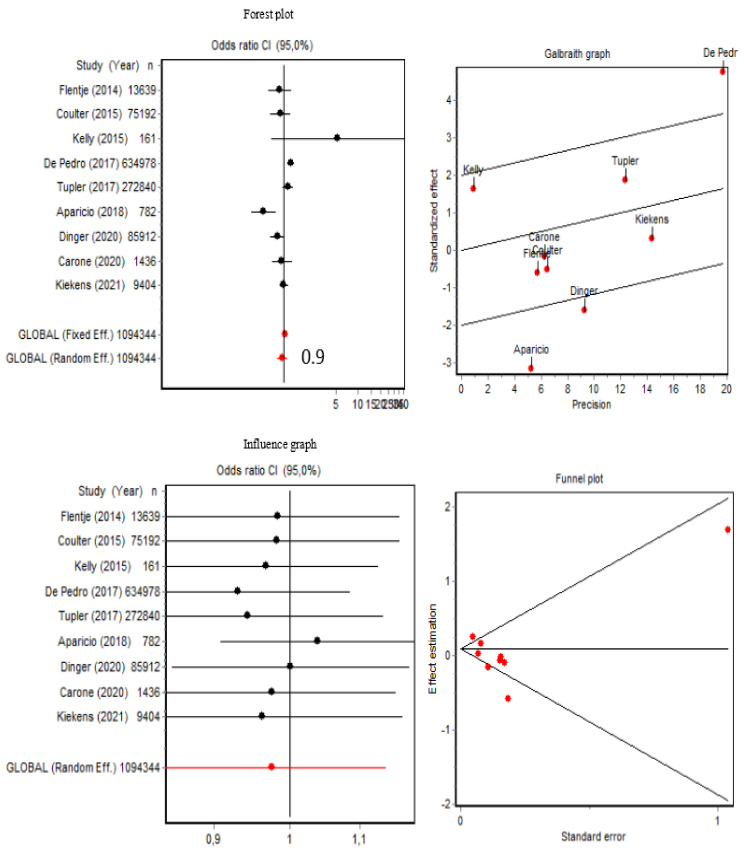
Forest plot, Galbraith graphic, influence graphic, and funnel plot for current alcohol use.

**Figure 4 brainsci-12-00366-f004:**
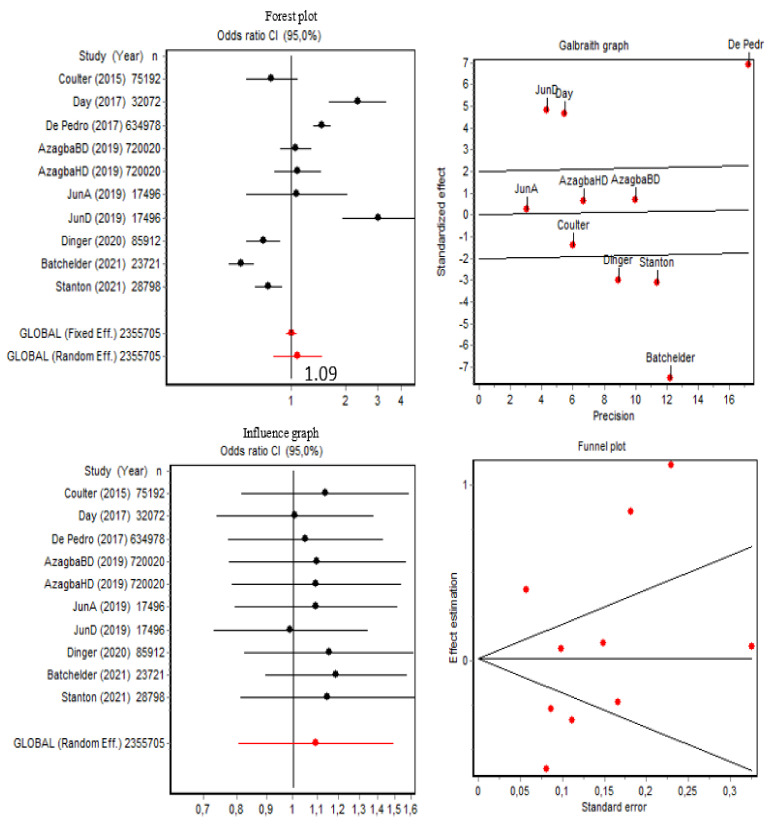
Forest plot, Galbraith graphic, influence graphic, and funnel plot for current alcohol use disorder. Abbreviations: BD, binge drinking; HD, heavy drinking; A, abuse; D, dependence.

**Figure 5 brainsci-12-00366-f005:**
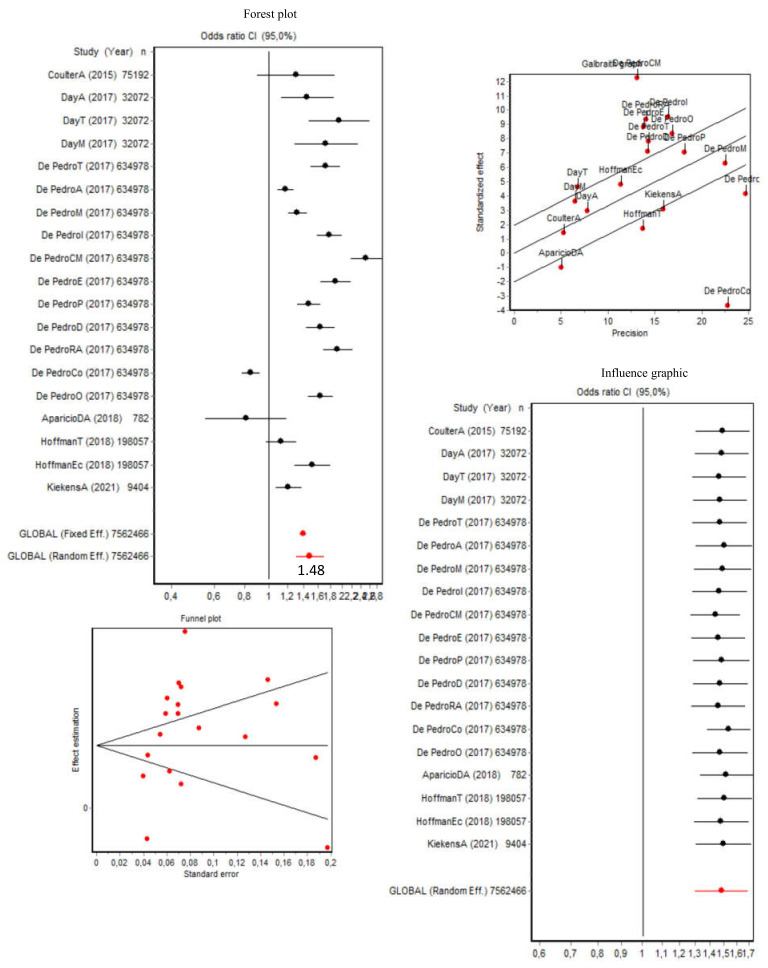
Forest plot, Galbraith graphic, influence graphic, and funnel plot for lifetime substance use (including tobacco, alcohol, and other drugs). Abbreviations: A, alcohol; T, tobacco; M, marijuana; I, inhalants; CM, cocaine/methamphetamine; E, ecstasy; P, prescription painkillers; D, diet pills; RA, Ritalin or Adderall; Co, cold medicine; O, other drugs; DA, drugs and alcohol; EC, e-cigarettes.

**Figure 6 brainsci-12-00366-f006:**
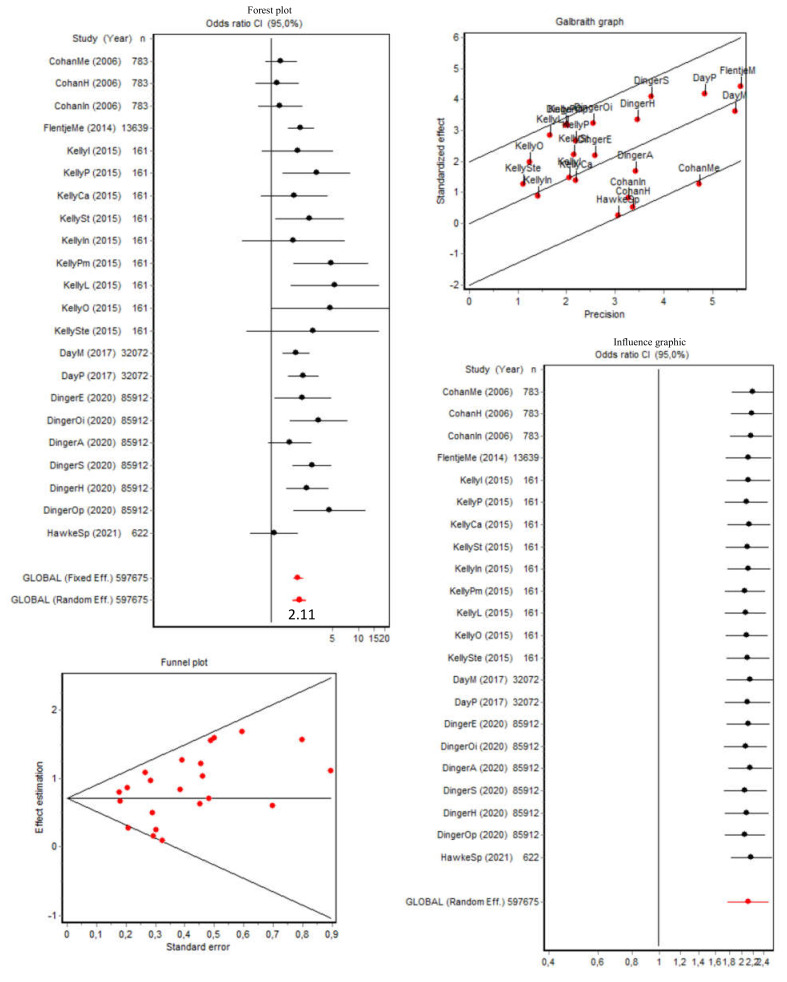
Forest plot, Galbraith graphic, influence graphic, and funnel plot for the current use of specific substances (excluding tobacco and alcohol) with subgroup analysis. Abbreviations: Me, methamphetamines; H, heroin; In: inhalants; P, polysubstance; Ca, cannabis; St, stimulants; Pm, prescription medications; L, LSD; O, opiates; Ste, steroids; M, marijuana; E, ecstasy; Oi: other illegal drugs; A, amphetamines; S, sedatives; Sp, current substance use pre-COVID.

**Table 1 brainsci-12-00366-t001:** Studies included in this review [12,19,28,29,30,31,32,33,34,35,36,37,38,39,40,41,42,43,44,45].

1st Author Year	Country	Population	Age in Years	Sample Size	Substance and Use	NOS Quality
Aparicio-García 2018	Spain	Contacts of LGBT + associations	14–25	782	Drugs and alcohol LU, Tobacco CU, Alcohol CU, Drugs CU	7
Azagba 2019	USA	General population	>18	720,020	Tobacco CU, UD; Smokeless tobacco CU, UD; Alcohol UD (heavy and binge drinking)	8
Batchelder 2021	USA	Community health centre	Mean 31.5	23,721	Alcohol UD, Substance UD	5
Buchting 2016	USA	General population	>18	17,164	Tobacco CU	8
Carone 2020	USA	General population	>18	1436	Alcohol CU, Drugs CU	8
Cohan 2006	USA	Sex workers	17–76	783	Tobacco CU, Illicit drug CU, Methamphetamines CU, Heroin CU, Cocaine/crack CU, Injection drugs CU	8
Coulter 2015	USA	Postsecondary students	18–29	75,192	Alcohol LU, Alcohol CU, Alcohol UD (heavy drinking)	8
Day 2017	USA	Middle and high school students	10–18	32,072	Alcohol LU, UD (heavy drinking); Tobacco LU, CU; Marijuana LU, CU; Other drugs CU; Polysubstance CU	8
De Pedro 2017	USA	Middle and high school students	Not reported	634,978	Tobacco LU, CU; Alcohol LU, CU, UD; Marijuana LU, CU; Inhalants LU, CU; Cocaine/Methamphetamine LU; Ecstasy LU; Prescription painkillers LU, CU; Diet pills LU; Ritalin or Adderall LU; Cold Medicine LU; Other drugs LU, CU; 2 or more drugs CU	8
Dinger 2020	USA	College and university students	18–25	85,912	Cigarettes CU; E-cigarettes CU; Alcohol CU, UD (binge drinking); Marijuana CU; Amphetamine CU; Sedative CU; Hallucinogens CU; Opiate CU; Inhalant CU; MDMA (Ecstasy) CU; Other club drugs CU; Other illegal drugs CU	8
Flentje 2014	USA	Abuse treatment services clients	Mean 38.31	13,639	Alcohol CU, Cocaine CU, Heroin CU, Methamphetamine CU, Marijuana CU, Other drugs CU	8
Gamarel 2020	USA	Sexual and gender minorities	13–17	8243	Tobacco CU	7
Hawke 2021	Canada	Clinical and nonclinical	14–28	622	Substance CU (pre-COVID and intra-COVID)	6
Hoffman 2018	USA	Noninstitutionalised population	>18	198,057	Cigarettes LU, CU; E-cigarettes LU; CU; Nicotine UD (dependence)	8
Jun 2019	USA	General population	20–35	253,033	Nicotine UD (dependence), Alcohol UD (abuse and dependence), Drug UD (abuse and dependence)	8
Kelly 2015	Australia	LGBT festival goers	13–24	161	Alcohol CU, Tobacco CU, Any illicit drug CU, Poly-drug CU, Cannabis CU, Stimulants CU, Inhalants CU, Prescription medications CU, LSD CU, Opiates CU, Steroids CU	5
Kiekens 2021	USA	Sexual and gender minorities	13–17	9404	Alcohol CU, Alcohol LU	7
Stanton 2021	USA	Community health centre specialising in sexual and gender minorities	>18	28,798	Alcohol UD, Substance UD	6
Tamí-Maury 2015	USA	LGBT individuals participating in the Pride Parade and Festival	Mean 30	94	Tobacco CU	5
Tupler 2017	USA	College students	>17	272,840	Alcohol CU	7

Note: USA, United States of America; LGBT, Lesbian, Gay, Bisexual, and Transgender; LU, Lifetime use; CU, Current Use; UD, Use Disorder; NOS, Newcastle–Ottawa Scale.

**Table 2 brainsci-12-00366-t002:** Summary of the results.

DATA	N1	OR1	95% IC1	N2	OR2	95% IC2
Current tobacco use	14	1.65 *	1.37, 1.98	9	1.58 *	1.44, 1.73
Current tobacco use disorder	3	1.52	0.94, 2.45	2	1.29	0.79, 2.11
Current alcohol use	9	0.97	0.83, 1.14	6	1.03	0.93, 1.15
Current alcohol use disorder	10	1.09	0.80, 1.49	4	1.01	0.69, 1.28
Lifetime substance (all) use	19	1.48 *	1.30, 1.68	7	1.38 *	1.28, 1.49
Current substance use (excluding tobacco and alcohol)	5	1.12	0.58, 2.15	3	0.98	0.70, 1.37
Current use of specific substances (excluding tobacco and alcohol)	40	1.79 *	1.54, 2.59	22	2.11 *	1.77, 2.51
Current substance use disorder (excluding tobacco and alcohol)	4	1.53	0.91, 2.59	3	1.06	0.80, 1.40

Note: N = Number of data included in the meta-analysis. OR = Pooled odds ratio of transgender people compared to cisgender people. IC = Confidence interval. N1, OR1, and 95% IC1 refer to the first meta-analysis performed, while N2, OR2, and 95% IC2 refer to the meta-analysis performed after removing the data that most contributed to heterogeneity. Significant results are marked with an asterisk.

## Data Availability

Data from this study are available from the corresponding author upon request.

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
