# Peer review of "Substance Use in the Transgender Population: A Meta-Analysis"

_brainsci, 2022, doi:10.3390/brainsci12030366_

Round 1
Reviewer 1 Report
In this article (Substance use in the transgender population: a meta-analysis), the authors aimed to assess the relationship between identifying as transgender and substance use. The results of this study show that, compared to the cisgender population, transgender individuals had a higher probability of presenting tobacco and substance use, but they did not differ in terms of alcohol use or alcohol, tobacco and substance use disorder. The author concludes that considering the transgender population as consumers or as addicted may be a prejudice which perpetuates stigma. Moreover, it is of particular importance to implement policies against discrimination and stigmatization and to adapt treatment services so that they are inclusive of the LGBTQIA+ community.
The article is interesting and explores an important and timely topic.
However, there are some issues the authors should address:
- in the method section the author should expand “2.8. Sensitivity and subgroup analysis” explaining more in detail why and for which factors the sensitivity and subgroup analysis are conducted
- judgments and justification of results should be posted in the discussion section instead of in the results section. For example, “probably because of their small sample size” (line 194)
- the results are not immediately comprehensible, considering the many different classifications are considered at the same time. I suggest the authors to create a "summary of findings" table.
- It would be interesting to evaluate the impact of psychiatric comorbidities on illicit substance use, alcohol use disorder or tobacco use disorder as a possible moderator during the analysis. If the included papers in the meta-analysis do not provide enough data to allow this type of analysis, the authors should discuss more in detail the possible impact of psychiatric comorbidities on substance abuse in the discussion section (https://doi.org/10.1159/000090429).
- The authors should extend the discussion about current critical issues in substance abuse management, considering that there is a need for better integration of programs for alcohol and other substance disorders into the mental health system of care or more effective referral procedures between the two separate systems of care (http://dx.doi.org/10.1016/j.comppsych.2014.11.021)
Carrà G, Scioli R, Monti MC, Marinoni A. Severity profiles of substance-abusing patients in Italian community addiction facilities: influence of psychiatric concurrent disorders. Eur Addict Res. 2006;12(2):96-101. doi: 10.1159/000090429. PMID: 16543745.
Author Response
Thank you so much for your opinion. We believe that your contributions have made the article more understandable, especially the results section with their summary.
We have add: Specifically, the meta-analyses were repeated eliminating the articles that contributed the most to heterogeneity according to the Galbraith graph (lines 205-206). In addition, it has been included in limitations: The heterogeneity and sensitivity analysis and the low number of studies in some of the analysis and categories have not allowed the analysis of specific subgroups, but the influence of variables such as age, sample type and other identities is an open question (lines 756-759).
It is true that the place for judgments and justification of results is discussion. We have removed that phrase. However, as not all readers will be familiar with the program used and the interpretation of its graphs, we believe that including some of its interpretations in the results section helps their understanding.
We have created such a table and included it in the article as table 2 (lines 383-388). Indeed, in this way it is much more understandable for the readers. Thanks.
Unfortunately, the data of the articles included do not allow it, so we have included in the discussion: It is also interesting to consider the impact of psychiatric comorbidities on illicit substance use as a possible moderator. Most papers included in our analysis do not explore this relationship, but there is evidence that transgender people suffer a high prevalence of mental health disorders, highlighting anxiety [Bouman et al., 2017], depression, self-harm, and suicidal ideation [Witcomb et al., 2018]. Previous studies carried out in the cisgender population demonstrate the impact that psychiatric comorbidities have on substance abuse, increasing the severity of addiction and functional impairment [Carrà et al., 2006](lines 384-706).
We have included in the discussion: Although there are no specific studies focused on dual disorders in transgender individuals, the prevalence of psychiatric comorbidities in this population leads us to hypothesize that substance abuse could be modulated by these diseases; being more severe and requiring a comprehensive approach that integrates or improves coordination between substance abuse and mental health care systems [Carrà et al., 2006](lines 706-710).
Thanks again for your suggestions, they have been very helpful.
Reviewer 2 Report
Dear authors,
I am delighted for the opportunity to review your research review and meta-analysis on substance use in transgender population for Brain Sciences.
My individual comments are listed below.
Abstract:
- Please describe your basic methodological approach for meta-analysing the data (variables, model).
- What does “specific substances” mean? Which substances were considered?
- The abstract’s conclusion does not adequately relate to the presented results. Instead, different discussion points are raised without deeper elaboration. I would avoid such statements in the abstract, which can be extensively discussed in the discussion section of the review paper.
Introduction:
- Some details about terminology: I would rather use consistently the term “biological sex” instead of “anatomical sex”. Also, not all transgender people may wish to undergo hormonal and/or surgical transition.
- Next to people identifying themself as genderqueer and gender-fluid, there are intersex and asexual individuals.
- I am not fully convinced by your theoretical framework proposed in the introduction section. Your main argument addresses the minority stress theory which could lead to higher levels of substance use. However, some recent findings on this topic are missing (e.g., Connolly & Gilchrist, 2020), and language that is less deterministic may improve theory building (e.g., “The use of this substance constitutes a 60 avoidant coping skill used to escape from negative emotions”).
- The research objective is not clear and, in my view, not appropriate to be answered by meta-analysis (in general). For example, it seems to include multiple objectives: (1) something about the use of substances in transgender persons and (2) about SUD in transgender persons. However, it is unclear whether you were interested in the prevalence or frequency or risk compared to cisgender people or any other possible outcomes.
Methods:
- What is a “quantitative comparison study”? Please specify what needs to be compared in the study.
- Which substances were considered (inclusions criteria)? Was only substance use in general or also substance specific substance use of interest?
- You use very different terms: abuse, addiction, dependence but you do not define these terms. Please specify what is meant by these terms and how you defined it.
- Were studies searched within a certain time range or were all publications since the relevant databases existed searched?
- Please provide a definition of current substance use (related to which period?) and SUD (DSM, ICD? Or by screening instrument?).
- How were weights calculated and considered in the random-effects model you employed?
- Please provide some more information about the NOS. How many researchers performed quality assessment?Results:
- Figure 1 includes a minor typo: “2,097 excluded according to THEIR title and abstract”; moreover, it seems that some text is missing in “277 articles assessed for *full text screening*? “
- Please also specify what “specific substances” mean in Figure 1.
- Please check Table 2 to be self-explained (e.g., “age in years”, “general ??”, “Sample size” …). Explain all abbreviations in the notes.
- I suggest to include only the forest plots into the manuscript and to provide funnel plots etc. in the Supplement. Also, instead of just naming the pooled OR, it would be good to have some more information about this OR: What does it mean? E.g. “the pooled OR for current tobacco use was 1.65 in transgender person compared to cisgender persons. In other words, transgender persons were at 65% increased risk of reporting current tobacco use…”
- Also, I strongly recommend to add a brief descriptive summary for those outcomes where only very few studies were available. As you indicated in the results section, it does not make sense to summarise three or four studies in meta-analysis so you may wish to consider to provide just a brief description of studies.
- In general, it would be excellent, if the weights could be given in the forest plots (e.g., next to the sample size).
Discussion:
- With regard to a lack of difference in alcohol use: Given the high prevalence of alcohol use in general, lack of difference could be because of differences in drinking patterns. With regard to AUD, lack of difference could also be caused by different indicators of AUD. However, as you do not explain how AUD/SUD was defined, it is unclear whether this could really explain a lack of difference. (Although it could also be that there is really no difference between both groups.)
- Your search terms were restricted to substance use or drug use in general which may have led to the exclusion of some specific substances. This might be particularly relevant for alcohol and tobacco use, but also for, e.g., the use of opioids. This should be acknowledged in the limitation section.
- Also add that the search was only conducted in English, which is a good reason, why only English-language articles were found.
- It might be worth noting that all but two studies were from North America (and all but three from US).
- Based on your findings, you cannot deduce the conclusion, that they would support the emotional regulation strategy. You have not tested any model in your study, but only pooled risks between groups. This cannot explain underlying mechanisms.
I wish the authors best luck for their publication!
Author Response
Thank you so much for your opinion. We believe that your contributions have made the article much more understandable and the methodology and terms used have been greatly clarified. Thank you very much.
Abstract:
We have added in the abstract: These articles included data on current tobacco use, current tobacco use disorder, current alcohol use, current alcohol use disorder, lifetime substances (all) use, current substance use (excluding tobacco and alcohol ), current use of specific substances (excluding tobacco and alcohol and including cocaine, amphetamines, methamphetamines, ecstasy, stimulants, heroin, opiates, cannabis, marijuana, LSD, hallucinogen, steroids, inhalants, sedatives, Ritalin or Adderall, diet pills, cold medicine, prescription medications, polysubstance, other club drugs, and other illegal drugs), and current substance use disorder (excluding tobacco and alcohol). We used the ORs and their 95% CIs to state the association between identifying as transgender and those variables. The control reference category used in all cases was cisgender. We employed a random-effects model (lines 14-23).
Substances are specified in the added paragraph (lines 17-20). We think this makes it clearer. Thanks for the suggestion.
It is true that the summary of conclusions includes hypotheses to explain the results that are not derived directly from the data. We have tried to distinguish by means of verb tenses the conclusions derived from the data (The presence of substance use disorders did not differ between transgender and cisgender people) from those that are hypotheses (may/might be). To make it clearer, we've added: Hypothetically (line 31).
Introduction:
That is right, we have changed it in the introduction. Thank you.
We have added in the introduction: Some people prefer to view these congenital conditions as a matter to diversity and use the terms intersex o intersexuality instead. This term includes not only people whose gender identity differs from the sex assigned at birth, but also those whose reproduc-tive organs do not conform to what is traditionally designate as male or female. These terms encourage the conception of gender from a non-binary perspective of it. Transgender is an umbrella term preferred by many because it is more inclusive. The intersex concept must be differentiated from people who identify themselves as asexu-al, which represents those people who typically do not experience sexual attraction or want to pursue sexual relationships with other people (lines 44-58).
It is true that this article has been included in the discussion but not in the introduction. This reference has been added in the introduction with the phrase: There is evidence to support the minority stress model but, in the case of the transgender population, it might still be scarce, as much of the research has focused on transgender women with multiple intersectional disadvantages [Connolly & Gilchrist, 2020](lines 90-92).
Less deterministic language has been used: could constitute (line 84).
We have added in the introduction: Our objective was to group and analyse the results published to date in order to quan-tify the probability (pooled odds ratio) of presenting use and suffering from a sub-stance use disorder among transgender people compared to the cisgender population, in other words, to compare the prevalence of substance use and substance use disorder in transgender and cisgender people (lines 103-128).
It is true that the introduction does not include all the specific substances studied, since until we carried out the search and extracted the data from the studies, we did not know what they would be. For this reason, the specific data available, the variables on which meta-analyses are going to be carried out, are specified in results, specifically in the flowchart. In addition, we have included a table with the summary of the results at the end of results section to facilitate understanding. We hope this makes it clearer. Thanks.
Methods:
We have added: quantitative comparison studies regardless of the design, comparing the prevalence of substance use and disorders between transgender and cisgender subjects, whether they consider substances in general or specific substances (lines 142-144).
Both are considered, so it has been added: whether they consider substances in general or specific substances.
Summary measures section has been completed: We used the ORs and their 95% CIs to state the association between identifying as transgender and current substance use (if the subject currently uses the substance, if he/she has used it in the last 15-30 days), current substance use disorder (if the subject is diagnosed with a substance use disorder, including both abuse (the substance continues to be consumed despite the problems and negative consequences it causes) and dependence (substance use causing tolerance, withdrawal, and/or pattern of compulsive use)), and lifetime substance use (if the subject has ever consumed the substance throughout life)(lines 172-184).
Since the relevant databases existed searched. We have added: The search start year was not limited (line 137).
We have defined this in the added paragraph in lines 172-184.
The articles use different modes of evaluation and some do not specify it, for this reason, we have added in limitations the heterogeneity in the way of evaluating the type of consumption (lines 761-762).
Epidat weights the studies by the inverse of the variance. It has been specified in the text (line 186).
Two of the investigators independently performed the quality analysis, then compared the results until consensus was reached. We have specified it in the text (lines 194-195 and 860-861).
Results:
Right, we have corrected figure 1. Thanks.
We have done it in figure 1.
We have done it in table 1.
This is a good idea as it would make the manuscript more visually pleasing, but we think it would make it difficult for the reader to follow the results for heterogeneity, sensitivity and publication bias.
We have done it in the variables in which there are significant differences in prevalence between transgender and cisgender subjects (lines 243, 306, 348 and 358).
We have done it (lines 260-262, 325-328, and 367-369).
It is an excellent idea, but unfortunately Epidat's graphical output does not allow this variable to be included.
Discussion:
In response to previous suggestions, the definition of these diagnostic categories and the limitation that the variety in their evaluation supposes have been included. We hope this makes it clearer.
We have added in limitations: Likewise, the search terms used could have left studies referring to a specific substance out of the results (lines 766-778).
We have added in limitations: The fact that these terms were in English has probably influenced that all the studies were in that language and, except for two, the rest were carried out in North America (most of them in the United States)(lines 778-780).
We have specified it and added the word hypothetically (line 815).
Thank you very much again for your suggestions.
Reviewer 3 Report
Thank you for the opportunity to review Substance use in the transgender population: a meta-analysis. This meta-analysis aimed to examine alcohol, tobacco, and drug use, as well as substance use disorder status among individuals who identify as trans compared to individuals who identify as cisgender.
Strengths of the manuscript include the importance of the topic, use of PRISMA guidelines, and the authors’ efforts toward increasing inclusivity of the trans population through research.
Please find below my questions, concerns, and suggestions for each section of the manuscript. It is my hope that my feedback allows you to strengthen the quality of this work.
Introduction
Terminology and definitions used need revising.
When defining transgender, please delete “meaning that they identify with roles culturally assigned to the other sex” (line 31). This is incorrect and is instead describing transsexual, a term which only describes a subpopulation of the trans community. Further, please consider noting that transgender is an umbrella term which can include people who identify as transsexual. I think it is also worth mentioning that transgender is the preferred term by many because it is more inclusive.
When describing gender minorities (line 38), non-binary, agender, and two-spirit should also be listed.
When using the LGBTQIA+ acronym, consider using 2SLGBTQIA+ to be inclusive of Native Americans who are members of this community.
Transmasculine and transfeminine are umbrella terms for folks who were not assigned female at birth (AFAB) and identify as a feminine to some degree. The parentheticals “female-to-male” and “male-to-female” implies is more in line with the more rigid definition of transsexual. If referring to people who are AFAB and now identify as men/masculine, these are trans men. Please be clear and consistent with terminology throughout the manuscript. Please clarify which population you are referring to when providing prevalence rates.
Lines 47-48: “Transgender individuals can experience intense stigma along with social exclusion 47 and marginalization.” This needs a citation.
Lines 60-65: The statement about avoidant coping needs to be qualified. This statement, as written, suggests all alcohol use by trans folks is a coping mechanism to deal with stresses from transphobia and identity concealment. However, one of the references cited [19] demonstrates trans people reported both enhancement and social drinking motives (both of which are considered positive-reinforcement motives). This may especially the case when folks meet up with friends at 2SLGBTQIA+ affirming bars, which the author indicates is one of the few safe spaces for this community. The language in that sentence also needs to be qualified. For some individuals, these bars may be one of the few safe spaces, but this is not true for all trans people. Consider rephrasing “more difficult to escape recreational substance use” to “may increase recreational alcohol use.”
Methods
Figure 1
2,373 articles whose titles were …? Seems like a phrase is missing.
Why was outcome of quantity or frequency of consumption an exclusion criterion in the study? Could this data not be used to deduce current substance use?
Why was sex assigned at birth extracted (but never reported) but not trans identity? How was transgender operationalized? Are non-binary trans people in the samples included in the meta-analysis?
How was “current substance use” defined? Past-month, past-year, or something else?
Given the extreme heterogeneity in the type of samples, a mixed effect model would be more appropriate. This would allow for adjusting for age (since at least 29% of the sample is under 18), and certain samples have higher risks for illicit substance use (e.g., people engaging in sex work), which may vary across countries with different laws around substance use (and sex work). Minimally, sensitivity analyses on these subgroups are warranted.
Results
Lines 162-164: Table 1 shows the characteristics of the studies included in these meta analyses [12, 19, 27–44] which were all in English and included both sexes assigned at birth. What does “included both sexes assigned at birth” mean?
Please be more specific when categorizing a sample as “sex workers.” Are you referring to legal or illegal activities? This is an umbrella term for a very diverse population.
Please clarify what you mean when you say, “the inaccuracy of the data from Kelly (2015).” How are these data inaccurate?
Define intra-COVID substance use.
Figure 5 and 6 Galbraith plots are difficult to read due to overlap.
Please consider adding abbreviations to all studies (e.g., binge drinking BD after author names in forest plot and influence graph).
Please include abbreviations after each study to denote the type of outcome reported on the alcohol figures.
Why are heaving drinking and binge drinking used in alcohol use disorder analyses? This only assesses consumption. These analyses should be redone and only studies which examined alcohol use disorder criteria should be included.
The lifetime substance use analyses are less informative due to the inclusion of tobacco and alcohol. I would recommend excluding these substances since you have separate tobacco and alcohol analyses.
Given the extreme heterogeneity of the samples and types of drugs included in the substance use analyses, I have concerns about interpreting these findings, especially since quantity and frequency of use are not assessed.
Discussion
My major concern is the interpretation of the findings and the pathologizing of substance use behaviors without adequate empirical support. While I appreciate the authors’ attempt to destigmatize the trans community and stop misinformation, please use language that does not stigmatize people who use drugs. The following statement needs revising:
Lines 456-457: This indicates that considering the population of transgender individuals as consumers and addicts likely represents a stigma-generating prejudice.
The term “consumer” is not common to describe people who use drugs (which is the preferred term). “Addict” is an offensive and outdated term no longer used to describe people who use drugs. If these terms are used to demonstrate the stigma, please use quotation marks.
A major limitation of the current study is that a minimum of 29% of the sample were under the age of 18. Stating that the samples included “young people” is not explicit enough. While you also acknowledge the sample heterogeneity in this section: “The fact that several studies were in populations from sexual and gender minorities may have also influenced the results, meaning that the prevalences would have been more similar than if they had been compared to the general cisgender population,” this is insufficient when describing combining samples which range from middle school children to people who engage in sex work (i.e., people at much higher risk for illicit substance use, especially amphetamines and opioids).
Lines 331-336: However, no differences were found in this study between transgender and cisgender people in terms of general substance use or substance use disorder. Nonetheless, these analyses included only a small number of studies, meaning that our confidence in the reliability of these results was low. Above all, previous studies 334 showed that transgender people were more likely to suffer from a substance use disorder, especially with amphetamines, cocaine, or cannabis [53, 58, 59].
Why would you have less confidence in the results of a meta-analysis than the results of 3 singular studies? “Above all” is strong language and is not consistent with your findings. Your results indicate that trans people may be more likely to use tobacco and substances but are not more likely to have substance use disorders, yet the discussion focuses on the potential roles of stigma and equitable. treatment.
Overall, the discussion seems to focus on the work of others, rather than the current work. Many of the studies discussed were not included in the meta-analysis.
The phrasing makes it seem like the authors are discussing their own work, rather than the work of others. For example:
Line 283: “Specifically, 64% reported having used tobacco, 23% had perceived their use as problematic at some point in their lives, and 13% believed their use was currently problematic.”
Additionally, what is the denominator for these percentages?
Line 302: Not only the prevalence of consumption, but also the age of consumption onset was earlier in the transgender group, often in the early stages of adolescence.
This sounds like a finding from the current study, and age of onset was an exclusion criterion for this study. Why is this being discussed?
Lines 315-319: They noted that having experienced a transphobic assault, homelessness, or involvement in sex work were common risk factors for cocaine and amphetamine use [53]. The use of amphetamines was more frequent in transgender people with low levels of education living in urban environments, and was associated with excessive alcohol consumption, symptoms of depression, reports of having suffered childhood emotional abuse, and the feeling that their environment did not support their identity [54].
Source 54 is not in the meta-analysis. As the reader, I am confused about why risk factors for amphetamine use are being discussed. I am also confused as to why homelessness and sex work are being discussed. While rates of these experiences are higher among trans compared to cisgender populations, this is not an accurate representation of the average trans person who uses drugs and/or alcohol. I have concerns that focusing on these relatively rare risk factors is potentially stigma-generating, especially since the authors acknowledge that trans people are often considered “addicts.”
Line 292: “these situations are more likely to occur in the group of transgender women.” To what group of transgender women are you referring? Same for “male cisgender group” in line 297. The more accurate terms are cisgender women and men (not male cisgender).
Line 300: Please use “compared to” rather than “versus” when describing population comparisons. This kind of language is indicative of othering.
Lines 350-351: “Moreover, the risk of violence victimisation in transgender students has also been significantly associated with substance misuse, especially in boys”
Who are “boys” in this context? Are you referring to children/adolescents assigned male at birth? Please be clear and consistent with terminology.
Author Response
Thank you so much for your opinion. We believe that, thanks to your contributions, the article has a more inclusive approach and is more respectful of diversity and gender perspective. Thank you very much.
Introduction:
We have removed that phrase (line 41) and added: Transgender is an umbrella term preferred by many because it is more inclusive (line 55).
We have included it in lines 61-62.
We have changed it throughout the text to make it more inclusive (lines 35, 76, 80, 86, 796 and 840). Thanks.
We have specified that the figures refer to transsexuality (line 68) and have eliminated the terms transfeminine and transmasculine.
We have added the reference (line 71).
We have modified it, being like this: For some subjects, the use of this substance could constitute an avoidant coping skill used to escape from negative emotions [16–20] and mitigate the stress produced by structural and internalised transphobia as well as identity concealment [15]. Bars with an 2SLGBTQIA+ atmosphere constitute one of the few spaces perceived by some subjects of this community as safe to meet and socialise without fear of discrimination, which could increase recreational substance use [21] because these places constitute environments that normalise consumption [22].
Methods:
Right, we have completed Figure 1. Thank you very much.
The aim of the study was to group and analyse the results published to date in order to quantify the probability (pooled odds ratio) of presenting use and suffering from a substance use disorder among transgender people compared to the cisgender population, in other words, to compare the prevalence of substance use and substance use disorder in transgender and cisgender people (lines 103-128).The data we needed was the odds ratio of transgender subjects compared to cisgender subjects (or the number of total trans and cis subjects and the number of trans and cis subjects who had substance use or substance use disorder to calculate it). When these data could be deduced, the article was included, but we excluded those where they could not be deduced, for example, those that only reported the average quantity or frequency of each group.
Sex assigned at birth was extracted because the first studies began in trans women (sex assigned at birth: male), in case this biased the results. However, all studies included in the study include both sexes assigned at birth. We have reported it in the article (lines 231-232). For the purposes of this study, we focus on cisgender and transgender identities.
Transgender was operationalized as individuals who identified themselves as transgender (line 145). If the studies differentiated between transgender and non-binary, only the group that identified as transgender was included (lines 154-155).
We have specified: if the subject has used it in the last 15-30 days (line 173-180).
It is true, but unfortunately the program used only allows a fixed or random effects model. On the other hand, the studies that introduce more heterogeneity are in turn heterogeneous in age and eliminating them does not significantly change the pooled odds ratio. We only have one study specifically in sex workers, all but two of the studies are in North American samples and all but three are in samples from the United States. Sensitivity analyzes indicate that removing any of the studies does not substantially change the result. And, in several of the analyses, the number of data is already quite limited, so creating subgroups would prevent having enough sample to reliably meta-analyse. For this reason, we have not carried out specific subgroup analyses, but we consider this to be a very important question to carry out when more research is available in this field. We include this issue in limitations: The heterogeneity and sensitivity analysis and the low number of studies in some of the analysis and categories have not allowed the analysis of specific subgroups, but the influence of variables such as age, sample type and other identities is an open question (lines 759-762).
We have added: male and female (line 232).
We have used the term used in the original article, where they do not refer to legal or illegal activities but: “We defined sex work as the provision of sexual services or performances in exchange for money or goods of economic value including but not limited to drugs, housing, and food”.
Because of the width of the confidence interval (line 265).
It is the term used by the original article, which distinguishes between use before the pandemic and use during the (current) pandemic.
It is true, but unfortunately the program used does not allow modifying the output, that is why we have explained the results of these plots in the text (lines 308-310 and 351-355).
We have included these abbreviations in the analyzes that included different substances for greater clarity, with respect to the rest of the analyses, this information is contained in Table 1.
We are sorry to disagree but, for example, in Azagba et al. (2019), binge drinking was defined as having five or more drinks on one occasion for males and as having four or more drinks on one occasion for females. Heavy drinking was defined for males as having more than 14 drinks per week and for females as having more than 7 drinks per week. A consumption of this entity is practically impossible not to cause some negative consequence. Since we have included abuse (the substance is consumed despite the problems and negative consequences it causes)(lines 181-182) in substance use disorder, we have included binge drinking and heavy drinking in this category. We are aware that it may be a very strict criterion, which is why we commented on it in the discussion: A debatable issue is that we have considered binge and heavy drinking as abuse and therefore we have included it in alcohol use disorder. We have done this because we consider that a consumption of this entity is practically impossible not to cause some negative consequence, so we differentiate it from the use of alcohol, which would be a moderate consumption limiting intake to 2 drinks or less in a day for men and 1 drink or less in a day for women (lines 744-749).
We have analyzed alcohol and tobacco separately because they were the individual substances in which there were more studies. The ideal would be to have enough studies of each of the substances. However, the interest of this lifetime analysis was to know the probability of having consumed some substance throughout life, for this reason all the substances on which there were data have been included. We have commented on this issue in limitations: The lifetime substance use analysis could be less informative due to the inclusion of tobacco and alcohol; however, the sensitivity analysis indicates that excluding these studies would not substantially change the result (lines 787-790).
You are absolutely right, that is why we have commented on it in limitations (758-762) and we include the phrase: Another limitation is that the quantity and frequency of use are not assessed (lines 790-791).
Discussion:
We have changed consumer to people who use drugs and put quotation marks around the word “addicts” to show the stigma (line 813). Thanks a lot for the suggestion.
We have modified the limitations section to make these issues more explicit: As limitations of this study, the main is the heterogeneity between the studies we included with respect to their populations (students, general population, sexual and gender minorities, sexual workers), cohort age (a minimum of 29% of the sample were under the age of 18), consumption type (current use, lifetime use, abuse, and dependence) and how to evaluate it, and substances evaluated (lines 758-762).
We have changed the paragraph: These results contradict previous studies showing that transgender people were more likely to suffer from a substance use disorder, specifically with amphetamines, cocaine, or cannabis [52, 56, 57]. Nonetheless, our analyses included a small number of studies, therefore, the results must be interpreted with caution (lines 456-459).
The meta-analysis synthesizes the results of the included studies, so we compared it with other studies not included to give a broader view. We have modified the points indicated below to differentiate our study and the included studies from others.
The paragraph has been modified: In relation to tobacco, it has been found that the use of cigarettes, cigars, or e-cigarettes is higher among transgender individuals compared to their cisgender peers [31]. In a study that included 350 transgender people (…) (lines 393-398).
The paragraph has been modified: In our study the prevalence of consumption is higher, while in other study the age of consumption onset was earlier in the transgender group, often in the early stages of adolescence [50] (lines 415-417).
We have eliminated this paragraph, which could certainly be stigmatizing. Thank you.
We have added: compared to transgender men (line 405).
We have changed the word (line 410).
We have changed the word (line 414).
We have changed the text quoting verbatim the words of the referenced article: especially in students who identify themselves as transgender or male (line 475).
Thanks again for your comments to strengthen the article.
Round 2
Reviewer 2 Report
The authors have satisfactorily addressed my comments.
Author Response
We are glad of your opinion. Thank you very much again for your helpful suggestions.
Reviewer 3 Report
Thank you for your thoughtful revisions and your willingness to use more inclusive language (e.g., use of Two-Spirited).
Please use "transgender" throughout rather than "transexual"/"transexuality" interchangeably. Gender and sexual orientation are distinct, and the term transexuality is a misnomer.
The discussion still focuses on substance misuse/abuse and advocates for inclusive substance treatment. While I agree this is important, your findings don't support this population having higher odds of substance use disorders. Perhaps inclusive prevention is more appropriate, since these folks are only more likely to use substances, especially early on (since much of the sample were middle/high schoolers).
Again, I would like to express my appreciation for the authors' commitment to inclusion and pursuing this important work.
Author Response
We have removed the definition of transsexual in line 41 and we have placed it in the only place in the text where we have kept this term because the referenced study refers specifically to it (lines 99-101). We have used "transgender" throughout the text (line 72, 454).
We have modified the discussion to include this important precision: This is especially important for prevention, as our findings show that transgender people do not have higher odds of substance use disorder, but rather are more likely to use substances, especially at younger ages (since much of the sample were middle/high students). It is critical to address addiction prevention in relation to sexual identity, lived experiences, related stressors, social and cultural contexts [64], and risky sexual behaviours [56]. To do this, anti-discrimination and stigmatisation policies must be implemented, and substance abuse prevention and treatment services should be adapted to be inclusive of the 2SLGBTQIA+ community (lines 834-853).
We have also added the word prevention in the abstract (line 34) and in the conclusions (line 870).
Many thanks again to you for your valuable input.